# Changes in the tumor microenvironment and outcome for TME-targeting therapy in glioblastoma: A pilot study

Sehar Ali[1], Thaiz F. Borin[1], Raziye Piranlioglu[1], Roxan Ara[1], Iryna Lebedyeva[2], Kartik Angara[3], Bhagelu R. Achyut[4], Ali Syed Arbab[1]*, Mohammad H. Rashid[1,5]*

**1** Laboratory of Tumor Angiogenesis Initiative, Georgia Cancer Center, Augusta University, Augusta, Georgia, United States of America, **2** Department of Chemistry and Physics, Augusta University, Augusta, Georgia, United States of America, **3** Department of Pediatrics and Human Development, Michigan State University, Grand Rapids, Michigan, United States of America, **4** Winship Cancer Institute, Emory University, Atlanta, Georgia, United States of America, **5** Department of Neurology, Cedars-Sinai Medical Center, Los Angeles, California, United States of America

* aarbab@augusta.edu (ASA); MohammadHarun.Rashid@cshs.org (MHR)

**Data Availability Statement:** All relevant data are within the manuscript and its Supporting information files.

## Abstract

Glioblastoma (GBM) is a hypervascular and aggressive primary malignant tumor of the central nervous system. Recent investigations showed that traditional therapies along with anti-angiogenic therapies failed due to the development of post-therapy resistance and recurrence. Previous investigations showed that there were changes in the cellular and metabolic compositions in the tumor microenvironment (TME). It can be said that tumor cell-directed therapies are ineffective and rethinking is needed how to treat GBM. It is hypothesized that the composition of TME-associated cells will be different based on the therapy and therapeutic agents, and TME-targeting therapy will be better to decrease recurrence and improve survival. Therefore, the purpose of this study is to determine the changes in the TME in respect of T-cell population, M1 and M2 macrophage polarization status, and MDSC population following different treatments in a syngeneic model of GBM. In addition to these parameters, tumor growth and survival were also studied following different treatments. The results showed that changes in the TME-associated cells were dependent on the therapeutic agents, and the TME-targeting therapy improved the survival of the GBM bearing animals. The current GBM therapies should be revisited to add agents to prevent the accumulation of bone marrow-derived cells in the TME or to prevent the effect of immune-suppressive myeloid cells in causing alternative neovascularization, the revival of glioma stem cells, and recurrence. Instead of concurrent therapy, a sequential strategy would be better to target TME-associated cells.

## Introduction

Even with current treatment strategies and the addition of immunotherapies or antiangiogenic therapies (as adjuvant), the prognosis of glioblastoma (GBM) is dismal [1–3]. GBM is a very hypervascular and invasive malignant tumor. So much so that, current treatments consisting

**Funding:** This study was supported by the Georgia Cancer Center startup fund and intramural grant program at Augusta University to Ali S. Arbab (ASA). The funders had no role in study design, data collection and analysis, decision to publish, or preparation of the manuscript.

**Competing interests:** The authors have declared that no competing interests exist.

of surgery, radiation, and chemotherapies with or without adjuvant still show no hope to patients [4–6]. Interestingly, recent investigations demonstrated that traditional therapies along with newer antiangiogenic therapies (as adjuvant) are changing the cellular as well as the metabolic compositions of the tumor microenvironment (**TME**) tremendously [7–11]. Studies have shown the role of N-hydroxy-N'-(4-butyl-2 methylphenyl) formamidine (HET0016), a highly selective inhibitor of 20-hydroxy arachidonic acid (20-HETE) synthesis (arachidonic acid metabolites) involving enzymes of the CYP4A and CYP4F families, in inhibiting tumor angiogenesis, proliferation, migration, and regulation of CD133+/CD34+ EPCs [12–14]. HET0016 was able to inhibit angiogenic responses to several growth factors as well as angiogenesis in gliosarcoma and the cornea induced by implanted human U251 GBM cells [15, 16]. Recent studies showed increased survival of animals bearing GBM following treatment with HET0016 [8]. Therefore, newer treatment strategies targeting TME including metabolites should be considered along with targeting tumor cells in GBM.

The TME is composed of tumor cells, stromal cells, cells from the bone marrow, and the extracellular matrix [17]. Except for a few cell types, such as normal epithelial cells, myoepithelial cells, dendritic cells, M1 macrophages, N1 neutrophils, and CD8 T-cells, most of the stromal and bone marrow-derived cells promote tumor growth and invasion [10, 11, 18–20]. Platelets have also been shown to promote tumor growth [21–24]. Therefore, it is imperative to include agents for targeting tumor-associated cells in the current standard regimen of therapies for malignant tumors such as GBM. However, there have been limited investigations done to understand the changes in the TME following standard as well as experimental therapies in GBM.

Tumor induction and evolution is driven by the interplay between stromal and immune cells within the TME. Tumor-associated macrophages (TAM), a critical component of the TME, have a differential function in respect to tumor growth and invasion [25–27]. TAM recruitment, localization, and phenotypes are regulated by the tumor-secreted factors at the hypoxic areas of the tumor [28, 29]. Depending on the stimuli, macrophages undergo a series of functional reprogramming as described by two different polarization states, known as M1 and M2 [29, 30]. Phenotypically, M1 macrophages express high levels of major histocompatibility complex class II (MHC II), the CD68 marker, and co-stimulatory molecules CD80 and CD86. On the other hand, M2 macrophages express high levels of MHC II, CD163, CD206/MRC1, Arg-1 (mouse only), and others. In the TME, classically activated macrophages, also known as M1 macrophages, are activated by tumor-derived cytokines such as granulocyte monocyte colony-stimulating factor (GM-CSF), interferon-γ, and tumor necrosis factor (TNF). These M1 macrophages play an important role as inducer and effector cells in polarized type 1 helper T cell (Th1) responses. These Th1 cells drive cellular immunity to eliminate cancerous cells. To accomplish Th1 activation, M1 macrophages produce high amounts of IL-12 and IL-23, and low amounts of IL-10, reactive oxygen and nitrogen species, and IL-1β, TNF, and IL-6 inflammatory cytokines [30, 31]. M1 macrophages also release anti-tumor chemokines and chemokines such as CXCL-9 and CXCL-10 that attract Th1 cells, [32–34]. Th1 cells drive cellular immunity to eliminate cancerous cells. On the other hand, M2-polarized macrophages, also known as alternatively activated macrophages are induced by IL-4, IL-13, IL-21, and IL-33 cytokines in the TME [35, 36]. M2 macrophages release high levels of IL-10 and, transforming growth factor-beta (TGF-β) and low levels of IL-12 and IL-23 (type 2 cytokines). M2 macrophages also produce CCL-17, CCL-22, and CCL-24 chemokines that regulate the recruitment of Tregs, Th2, eosinophils, and basophils (type-2 pathway) in tumors [32, 34]. The Th2 response is associated with the anti-inflammatory and immunosuppressive microenvironment, which promotes tumor growth.

Several chemokines such as macrophage colony-stimulating factor-1 (MCSF/CSF1) and monocyte chemotactic protein-1 (MCP1/CCL2) are known to contribute to the recruitment of

TAMs to the tumor [37, 38]. CSF1R expression has been reported on immunosuppressive myeloid cells and dendritic cells [39–41]. CSF1-CSF1R signaling regulates the survival, differentiation, and proliferation of monocytes and macrophages [42, 43], and has a critical role in angiogenesis and tumor progression [44, 45]. Previous studies indicated the involvement of bone marrow-derived myeloid cells in GBM development and CSF1R inhibition decreased cytokines that are involved in inflammation, angiogenesis, invasion, and proliferation, which eventually decreased GBM growth [10, 11, 46]. Previous studies showed decreased growth of GBM following treatment with CSF1R inhibitor (GW2580) [11].

Recent investigations including ours indicated the involvement of myeloid-derived suppressor cells (MDSCs) in the primary as well as metastatic TME [47–51]. MDSCs are a heterogeneous population of immature myeloid cells, generated from bone marrow hematopoietic precursor cells that fail to undergo terminal differentiation to mature monocytes or granulocytes. They are divided broadly into monocytic (CD11b+/Gr1+/Ly6C+) and granulocytic (CD11b+/Gr1+/Ly6G+) [52–54]. During tumor progression, MDSCs are greatly expanded and they exhibit remarkable immunosuppressive and tumorigenic activities. They are directly implicated in the escalation of tumor metastases by partaking in the epithelial-mesenchymal transition (EMT) and, tumor cell invasion, while also promoting angiogenesis and formation of the pre-metastatic niche [18, 48, 49]. MDSCs were demonstrated to promote tumor invasion and distal metastasis (although not in GBM) by two mechanisms: (i) increasing production of multiple matrix metalloproteinases (MMPs) that degrade the extra-cellular matrix and chemokines that establish a pre-metastatic milieu [55, 56], and (ii) merging with tumor cells [57, 58]. Different anti-depressant such as selective serotonin reuptake inhibitor (SSRI) has been used to treat GBM in an experimental setting but the detail of TME composition following therapy is not known [59–62].

From the above discussion, it is obvious that TME-associated bone marrow-derived cells are important in treatment resistance, invasion, and growth. In previous studies, the effect of HET0016, a potent inhibitor of arachidonic acid metabolites 20-HETE production, GW2580, a colony stimulation factor 1 receptor (CSF1R) antagonists, temozolomide (TMZ), Vatalanib, a VEGFR2 receptor tyrosine kinase inhibitor, SB225002, a CXCR2 inhibitor on the growth of GBM have been reported but systematic investigations were not performed regarding TME composition and survival [7, 8, 10, 20, 63]. Therefore, the purpose of this study is to determine the changes in the TME in respect of T-cell population, M1 and M2 macrophage polarization status, and MDSC population following different treatments in a syngeneic model of GBM. Transgenic animal models were also used, where CSF1R+ bone marrow-derived cells were conditionally knockout, treated with different agents, and the composition of TME was determined. In addition to these parameters, tumor growth and survival were also studied following different treatments in wild type and transgenic animals. In this study, the following agents were used: a drug that alters hydroxylase pathways of arachidonic acid metabolism (HET0016 and its different analogs), colony-stimulating factor 1 receptor (CSF1R) inhibitor (GW2580), anti-PD-1 (program cell death) antibody, CXCR2 receptor blockers (Navarixin and SB225002), temozolomide (TMZ), irradiation, (Vatalanib), fluoxetine, a selective serotonin reuptake inhibitor (SSRI), and conditional CSF1R knockout (KO) mice plus different treatments.

## Materials and methods

### Ethics statement

All the experiments were performed according to the National Institutes of Health (NIH) guidelines and regulations. The Institutional Animal Care and Use Committee (IACUC) of Augusta University (protocol #2014–0625) approved all the experimental procedures. All

animals were kept under standard barrier conditions at room temperature with exposure to light for 12 hours and dark for 12 hours. Food and water were offered ad libitum. Tumors were implanted orthotopically in animals with bodyweight between 20-22gm under ketamine (50mg/kg)-xylazine(10mg/kg) anesthesia (intraperitoneal injection). All animals received Buprenorphine SR (1.2mg/kg) subcutaneously to minimize post-surgical pain. The depth of anesthesia was checked by pinching skin or toe. The humane endpoint of the survival studies was the fulfillment of the criteria for euthanasia at the end of the survival studies (survival), which was by determining body weight (loss of more than 15% of baseline body weight), moribund, coma, paraplegia, inability to drink/eat. All animals were checked 2–3 times a week. A total of 114 animals was used for this study. 8 conditional knockout animals were used to confirm the depletion of CSF1R+ cells following poly-IC injection, 15 animals were used to determine the growth of tumor in wild type and KO animal, 40 animals were used for TME-associated cell determination by flow cytometry, and 51 wild type animals were used for survival studies. None of the animals that underwent TME analysis, analysis of CSF1R+ cell depletion, tumor growth in KO animals was dead prematurely. However, a few animals (5–6 animals) that were enrolled in survival studies were found dead on the day of scheduled euthanasia due to large intracranial tumors, which did not impact the calculation of survival. All animals were treated with soft chow, apple, and subcutaneous fluid when they started signs of intracranial tumor-related symptoms, such as head tilting, ataxia, ruffled fur, loss of weight, paralysis, lethargy, and dehydration. The animals were humanely euthanized once the euthanasia criteria were achieved. All efforts were made to ameliorate the suffering of animals. CO2 (displacement rate of 30–70% of the chamber volume with CO2 per minute) with a secondary method (bilateral thoracotomy or collection of major organs) was used to euthanize animals for tissue collection. Death was confirmed by established criteria of lack of breathing, lack of corneal reflex, lack of response to a firm toe pinch, and rigor mortis.

## Materials

HPßCD (2-hydroxy Propyl-β-Cyclodextrin) was purchased from Sigma-Aldrich (St. Louis, MO), cell culture media was from Thermo Scientific (Waltham, MA), and fetal bovine serum was purchased from Hyclone (Logan, Utah). HET0016 was made by Dr. Levedyeva in the Department of Chemistry, Augusta University with a purity of more than 97%, and was prepared for animal treatment according to our previously described method [8]. Additional information for HET0016 synthesis strategies is provided in the S1 File. Cell culture grade DMSO was purchased from Fischer Scientific (Pittsburg, PA). the complex of HET0016 plus HPßCD was made as per previous publication [8]. VEGFR2 tyrosine kinase inhibitor (Vatalanib) and colony-stimulating factor 1 receptor (CSF1R) inhibitor (GW2580) were purchased from LC Laboratories, Woburn, MA. SB225002 (CXCR2 inhibitor) was purchased from Selleckchem, Houston, TX. Navarixin was purchased from MedKoo bioscience Inc, Morrisville, NC. All flow antibodies are from Bio Legend, San Diego, CA. All antibodies for western blotting, immunohistochemistry, and immunofluorescence were purchased from Santa Cruz (total-CXCR2 and anti-GAPDH), R&D Systems (anti-hCXCR2), Thermo Scientific (anti-Laminin), and Sigma Aldrich (β-actin and FITC-conjugated tomato lectin). All culture media were purchased from Corning and GE Healthcare Life Sciences.

## Tumor cells and orthotopic animal model of GBM

To determine the *in vivo* effect of different treatments, orthotopic GBM models using syngeneic GL261 cells in wild type and CSF1R conditional knockout C57BL/6 mice (Jackson Laboratory) were prepared. The detailed methods are published previously [8, 10, 11, 64]. In short,

luciferase positive GL261 cells were grown in standard growth media (RPMI-1640 plus 10% FBS) and collected in serum-free media on the day of implantation. After preparation and drilling a hole at 2.25 mm to the right and 2 mm posterior to the bregma, taking care not to penetrate the dura, a 10 μL Hamilton syringe with a 26G-needle containing tumor cells (10,000) in a volume of 3 μl was lowered to a depth of 4 mm and then raised to a depth of 3 mm. During and after the injection, a careful note was made for any reflux from the injection site. The needle was withdrawn 1 mm at a time in a stepwise manner 2–3 minutes after completing the injection. The surgical hole was sealed with bone wax. Finally, the skull was swabbed with betadine before suturing the skin [65–67]. There were at least three animals in each group of treatment. Tumor growth was determined by optical imaging (bioluminescence imaging after injecting luciferin) on days 8, 15, and 22. For flow cytometry of tumor-associated cells, animals were euthanized on day 22 after the last optical imaging. Both male and female animals were used. For the treatment with Navarixin, TMZ+Navarixin, and corresponding control, athymic nude mice were used to create GL261 orthotopic GBM. We received GL261 mouse GBM cells from the repository of NCI. The cells are routinely tested for Mycoplasma and are treated with standard antibiotics during culture. The GL261 cell line was authenticated using short tandem repeat (STR) profiling by IDEXX BioResearch, MO, USA (Case # 23709–2016).

## Treatments

All treatments were started on day 8 following tumor implantation and continued for two weeks. List of treatments/agents, mechanisms of actions and the rationale behind combination therapies are shown in Table 1. The following treatment groups were used to determine the TME associated T-cells, different macrophages, MDSCs present by flow cytometry; 1) vehicle, 2) HET0016 complexed with HPßCD at 10mg/kg/day for 5 days/week, intravenous (IV), 3) GW2580, 160mg/kg/day 3day/week, oral, 4) temozolomide (TMZ) 50mg/kg/day, 3days/week, oral, 5) Vatalanib 50mg/kg/day, 5 days/week, oral, 6) Navarixin, 10mg/kg/day, 5 days/week, intraperitoneal (IP), 7) anti-PD-1 antibody, 200μg/dose, 2 doses/week, IP, 8) image-guided radiation therapy, 10Gy/dose/week for two weeks, 9) combined HET0016 plus GW2580, 10) combined HET0016 plus GW2580 plus anti- PD-1 antibody, 11) Fluoxetine 10 mg/kg/day, 3days/week, oral, and 12) combined TMZ plus fluoxetine.

## Making of a conditional knockout mouse model of bone marrow-derived CSF1R+ myeloid cells

The CSF1R flox female mouse (stock#021212) and MX1-Cre male mouse (stock# 003556) were purchased from the Jackson Laboratory. Mice were crossed to get Heterozygous CSF1R$_{flox/wt}$/MX1-Cre+ male mouse which was then backcrossed with CSF1R flox female (stock#021212, Jackson Laboratory). In another strategy, heterozygous CSF1R$_{flox/wt}$/MX1-Cre + male was mated with a heterozygous CSF1R$_{flox/wt}$/MX1-Cre+ female to achieve 25% of the progeny with homozygous CSF1R$_{flox/flox}$/MX1-Cre+ (knockout) genotype in bone marrow cells. Other progeny was wild-type CSF1R$_{wt/wt}$/MX-1-Cre+ (25%) and heterozygous CSF1R$_{flox/wt}$/MX-1-Cre+ (50%) genotypes. After repeated cross-breeding, a colony of CSF1R$_{flox/flox}$/Cre+ (knockout) have been generated. The transgenic animals showed no sign of abnormal weight loss, growth retardation, breeding, and survival. For genotyping, mouse genomic DNA was isolated from tail biopsies following digestion at 95˚C in lysis buffer containing 50 mM Tris-HCl (pH 8.0), 10 mM EDTA, 100 mM NaCl, 0.1% SDS, and 1mg/ml proteinase K, followed by heat inactivation. PCR was performed using the corresponding primer pairs. The samples were run in 2% (w/v) agarose gel and imaged by Biorad Gel Doc EQ System w/ Universal Hood II. Analysis of myeloid cells in the peripheral blood before and after

**Table 1. List of treatments and the rationale.**

| Treatments/drugs | Mechanisms of action | Rationale | Rationale for combination | Reference/PMID |
|---|---|---|---|---|
| Vatalanib | VEGFR2 receptor tyrosine kinase inhibitor | To block angiogenesis to decrease tumor growth | | [68] |
| Navarixin | CXCR2 antagonist | To block IL-8-CXCR2 interaction to reduce the formation of GBM stem cells, vascular mimicry, and decrease myeloid cells | | [69, 70] |
| SB225002 | CXCR2 antagonist | To block IL-8-CXCR2 interaction to reduce the formation of GBM stem cells, vascular mimicry, and decrease myeloid cells | | [71] |
| Temozolomide (TMZ) | DNA methylation | DNA damage and tumor cell death | | [72] |
| GW2580 (GW) | CSF1R inhibitor | Inhibits cFMS tyrosine kinase and inhibits the growth of myeloid cells | | [73] |
| Fluoxetine | A selective serotonin reuptake inhibitor (SSRI) | To decrease GBM cells | | [74] |
| HET0016 (HET) and its analog | The blocker of hydroxylase pathways of arachidonic acid metabolism causing decrease 20-HETE | Decrease tumor cell proliferation, decrease endothelial cell migration and proliferation, decrease inflammatory cascade and decrease neovascularization | | [8] |
| Irradiation | DNA damage, increase ROS | Decrease tumor cells | | [75] |
| TMZ+HET | | | DNA damage and tumor cell death and concomitantly inhibit the growth of tumor cell, endothelial cell. | [8] |
| HET+GW2580 | | | To inhibit the growth of tumor cell, endothelial cell, and myeloid cells | |
| HET+GW+anti-PD1 | | | To inhibit the growth of tumor cell, endothelial cell, myeloid cells and improve the level of cytotoxic T-cells | |
| CSF1R KD+anti-PD-1 | | | Deplete myeloid cells and improve the level of cytotoxic T-cells | |
| CSF1R KD+HET | | | To inhibit the growth of tumor cell, endothelial cell, and improve the level of cytotoxic T-cells | |
| CSF1R KD+anti-PD-1+HET | | | To inhibit the growth of tumor cell, endothelial cell, deplete myeloid cells and improve the level of cytotoxic T-cells | |

injection of polyinosinic-polycytidylic acid (poly-IC) showed bone marrow-specific depletion of CSF1R+ cells (Fig 1). These animals (male and female) were used to generate GL261 derived syngeneic GBM after depletion of bone marrow-derived myeloid cells and then treated with HET0016 or anti-PD-1 antibody alone or in combination or with CXCR2 antagonist SB225002 (10mg/kg/day 5 days/week, IP) for two weeks.

Primers used were (protocol from Jackson Laboratories) 26825 (5′-CAT GGC TGT GGC CTA GAG A-3′) and 16422 (5′-GGA CTA GCC ACC ATG TCT CC-3′) for CSF1R flox, Accession no: NC_000084 (REGION: 61245264..61245456). For Mx-Cre, Accession no/MGI ID: MGI:2176073, primers used were oIMR1084 (5′-GCGGTCTGGCAGTAAAAACTATC-3′) and oIMR1085 (5′-GTGAAACAGCATTGCTGTCACTT-3′). As an internal positive control, Accession no: NC_000069 REGION: 37176966..37177289, we also used primers oIMR7338 (5′-CTAGGCCACAGAATTGAAAGATCT-3′) and oIMR7339 (5′-GTAGGTGGAAATTCTAGC ATCATCC-3′) to assess a 324-bp product.

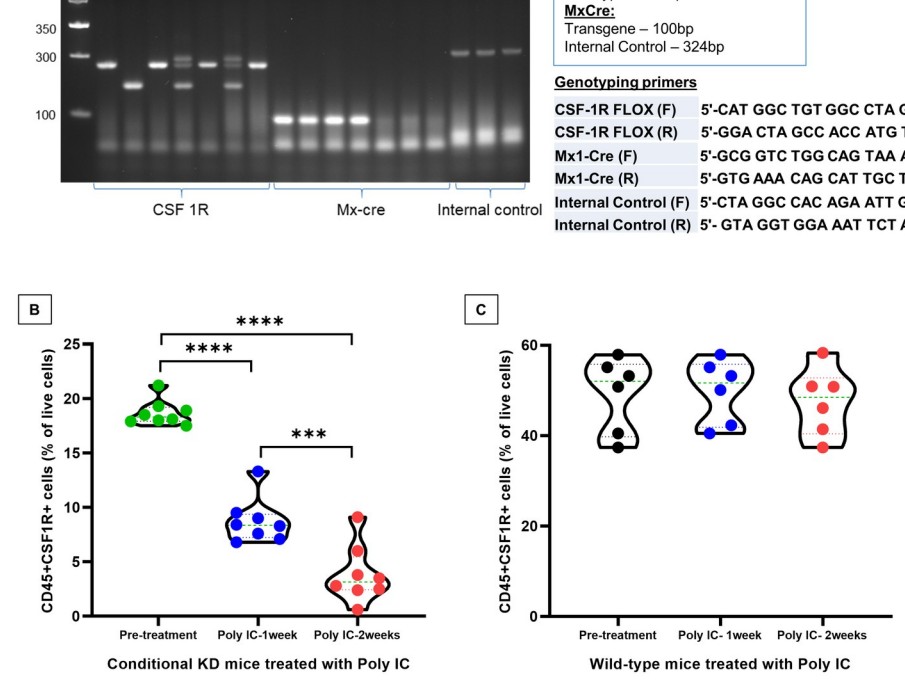

**Fig 1. CSF1R conditional knockout mouse and GBM development. (A)** Agarose gel electrophoresis showing homozygous CSF1R$^{flox/flox}$/MX1-Cre+ (knockout) genotype. **(B)** Violin plot showing flow-cytometric analysis of peripheral blood cells from conditional knock out mice showed a significant dose-dependent decrease in CD45+CSF1R + cells following two weeks of treatments with poly-IC. n = 8. **(C)** Violin plot showing flow-cytometric analysis of peripheral blood cells from wild type mice did not show any significant difference in CD45+CSF1R+ cells following two weeks of treatments with poly-IC. n = 6.

## Determination of bone marrow-derived cells in the TME

Following euthanasia on day 22 after tumor implantation, animals were perfused with ice-cold PBS and the right brain containing GBM was collected, passed through 40-micron mesh and a single-cell suspension was made and fixed with 3% paraformaldehyde for 5 minutes. After fixation cells were washed and re-suspended in PBS. Before adding panels of antibody cocktail, non-specific uptake of the antibody was blocked by adding the recommended blocker. The population of the following cells were determined by a flow cytometer from cells collected from tumors; CD45+/CD4+, CD45+/CD8+, CD45+/CD11B+/Gr1+/Ly6C+, CD45+/CD11B +/Gr1+/Ly6G+, CD45+/CD86+/CD80+, and CD45+/CD206+. The cells were labeled to detect the immune cell populations using fluorescence conjugated antibodies for CD3 (cat#100204), CD4 (cat#100512), CD8 (cat#100732), CD206 (cat#141708), F4/80 (cat#123116), CD25 (cat#101910), CD11b (cat#101208 & 101228), CD80 (cat#1047220), CD86(cat#105028), Gr1 (cat#108406), Ly6C(cat#128012), Ly6G(cat#127614), CD115(cat# 135526), and CD45 (cat#103108). All antibodies were mouse-specific (BioLegend), and the samples were acquired using the Accuri C6 flow cytometer (BD Biosciences). A minimum of 50,000 events was acquired. The findings were compared among all the treatment groups. Spleen from the conditional knockout animals was collected, made single-cell suspension and CSF1R+ cell populations were determined by flow cytometry.

## Determination of tumor growth

All animals that were not followed for survival were euthanized on day 22 to determine the TME associated cells also underwent optical imaging before treatment and at one and two weeks after treatments. The dose of luciferin and exposure time were kept identical for every animal at each time point. All animals underwent imaging following IP injection of luciferin (150mg/kg). Images were obtained from all animals on days 8, 15, and 22. Photon density (photon/sec/mm$^2$) was determined by drawing an irregular region of interest to cover the tumor area. The findings were compared among all the treatment groups.

## Determination of survival

Groups of animals (all were wild type animals) were also used to determine the survival following different TME targeted therapies. All animals were routinely observed 2–3 times a week to assess the wellbeing as well as body weight. The animals were followed up until they become moribund or fulfill the criteria for euthanasia as per the approved IACUC protocols. The findings were compared among all the treatment groups.

## Statistical analysis

Quantitative data were expressed as mean ± standard error of the mean (SEM) unless otherwise stated. For the flow-cytometric studies, ordinary one-way analysis of variance (ANOVA) followed by multiple comparisons using Dunnett's multiple comparisons test was used. For BLI (optical imaging) data, the general framework of analyses included two-way ANOVA followed by either Tukey's or Sidak's multiple comparisons. The survival of the animals following different treatments was analyzed. A Log-rank test (Mantel-Cox) was applied to determine the significance of differences among the groups. A P value of 0.05 was considered significant.

## Results

CSF1R-mediated signaling is indispensable for the proliferation, differentiation, chemotaxis, and survival of the myeloid cells [76]. CSF1R expression can be detected on myeloid cells within the TME such as TAMs, dendritic cells, neutrophils, and MDSCs [77]. As the presence of CSF1R+ cells in TME correlates with tumor progression, metastasis, and poor survival in various tumor types [78, 79], targeting CSF1R signaling in TME cells is an appealing strategy [77, 80]. In this study, CSF1R conditional knockout mouse was successfully developed. These conditional knock out mice showed homozygous CSF1R$^{flox/flox}$/MX1-Cre+ (knockout) genotype (Fig 1A). Compared to wild type mice, conditional knockout mice showed a significant dose-dependent decrease in CD45+CSF1R+ cells following two weeks of treatments with poly-IC. There was an almost 80% decrease of CSF1R+ cells in the peripheral blood (Fig 1B). Wild type mice treated with poly-IC did not show any significant difference in CD45+CSF1R+ cells (Fig 1C). Representative dot-plots are provided in the **S1 Fig in** S1 File.

In addition to CSF1R, which is a G-protein-coupled receptor, CXCR2 also can be found in neutrophils, TAMs, and MDSCs that regulates the homing of these cells in the TME [81, 82]. CXCL1 binds to CXCR2 and has been implicated in tumor growth, angiogenesis, and metastasis [69]. Kumar et al. noticed an upregulation of chemokines, most notably CXCL1 following the treatment of CSF1R-inhibitors that increased the recruitment of tumorigenic PMN-MDSCs in the TME [83]. They also demonstrated that while each inhibitor alone lacked an anti-tumor effect, combination treatment of CSF1R and CXCR2 inhibitors significantly reduced tumor growth by reducing the presence of both TAM and PMN-MDSC populations in the TME. To that end, the combined effect of CSF1R knockout and CXCR2 receptor

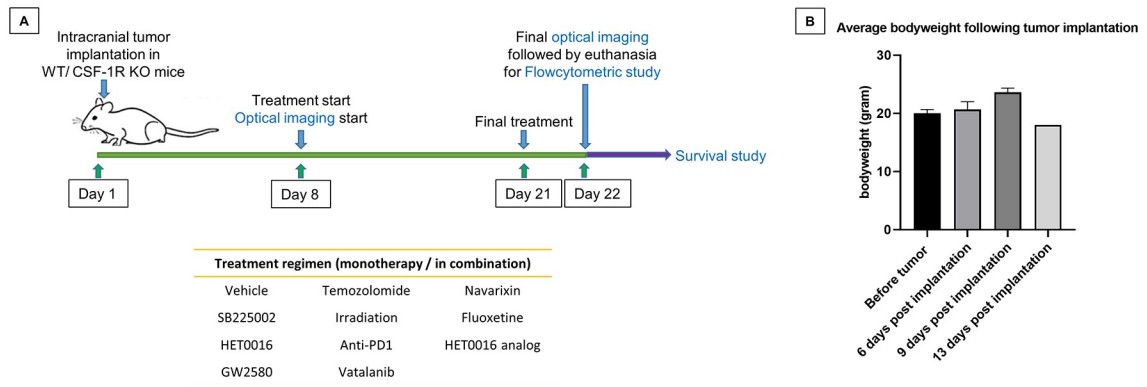

**Fig 2. (A)** Schematic representation of the study design and treatment schedule. **(B)** No significant difference in average bodyweights before and after the implantation of the tumor was observed over the period of 2 weeks.

blockers (SB225002) was also investigated. It is also to note that previous publication [7] indicated increased expression of CXCR2 in GBM cells in post-treated GBM causing recurrence and enhancement of alternative neovascularization. Therefore, CXCR2+ GBM cells were also targeted. Both wild type (control) and knockout mice (after two weeks of treatments with poly-IC) received intracranial implantation of syngeneic GL261 glioblastoma. Schematic representation of the study design and treatment schedule is shown in Fig 2A. There was no significant change in the average bodyweights of mice before and after the tumor implantation over the course of 2 weeks (Fig 2B).

On day 8 of tumor implantation, groups of animals received either vehicle or SB225002 for two weeks. All animals underwent optical imaging pre and post-treatment. Photon intensities were determined to measure tumor growth. Wild-type control animals showed significantly increased tumor growth (Fig 3A) which is indicated by a 10-fold increase in the photon intensity (Fig 3B). On the other hand, both wild type (control) treated with SB225002 and knockout mice showed significantly decreased tumor growth at week 3, indicating the involvement of CSF1R+ cells in the TME. Furthermore, a significant reduction of CSF1R+ cells distribution in the tumor and spleen of the conditional knockout mice compared to the wild type mice treated with vehicle and SB225002 (Fig 3C) was noted. Although there were no significant differences were observed, the addition of SB225002 further decreased the population of CSF1R+ cells both in the tumors and spleen. Representative dot-plots are provided in **S2 Fig in** S1 File.

It is also known that the CXCR2 antagonist can inhibit the function of myeloid cells by blocking the interaction of CXCR2 and IL-8 [84–86]. Tumor-associated CD45+CD11b+ (myeloid cells) and CD45+CD11b+CD206+ (TAMs) cells were determined following treatment with vehicle and SB225002 in brain tumor and spleen of the wild type and knockout animals. Both cell types were significantly decreased in knockout animals, prominently following the treatments (Fig 4A and 4B). Representative dot-plots are provided in **S3 and S4 Figs in** S1 File. T-cells and MDSC populations showed no significant difference between the treated and untreated wild type animals.

Both wild type and CSF1R knockout mice received different treatments that target tumor cells or tumor-associated cells. All treatments were for two weeks and the treatment was started on day 8 of orthotopic tumor implantation. On day 22 following the last optical imaging, animals were euthanized and the tumors were collected for flow cytometry to determine the population of T-cells (CD4, CD8), CD11b+ cells, macrophages (M1 and M2), and MDSCs (Ly6C and Ly6G). To our surprise, CD4, CD8, CD11b, Ly6C, Ly6G positive cells, and M1 and

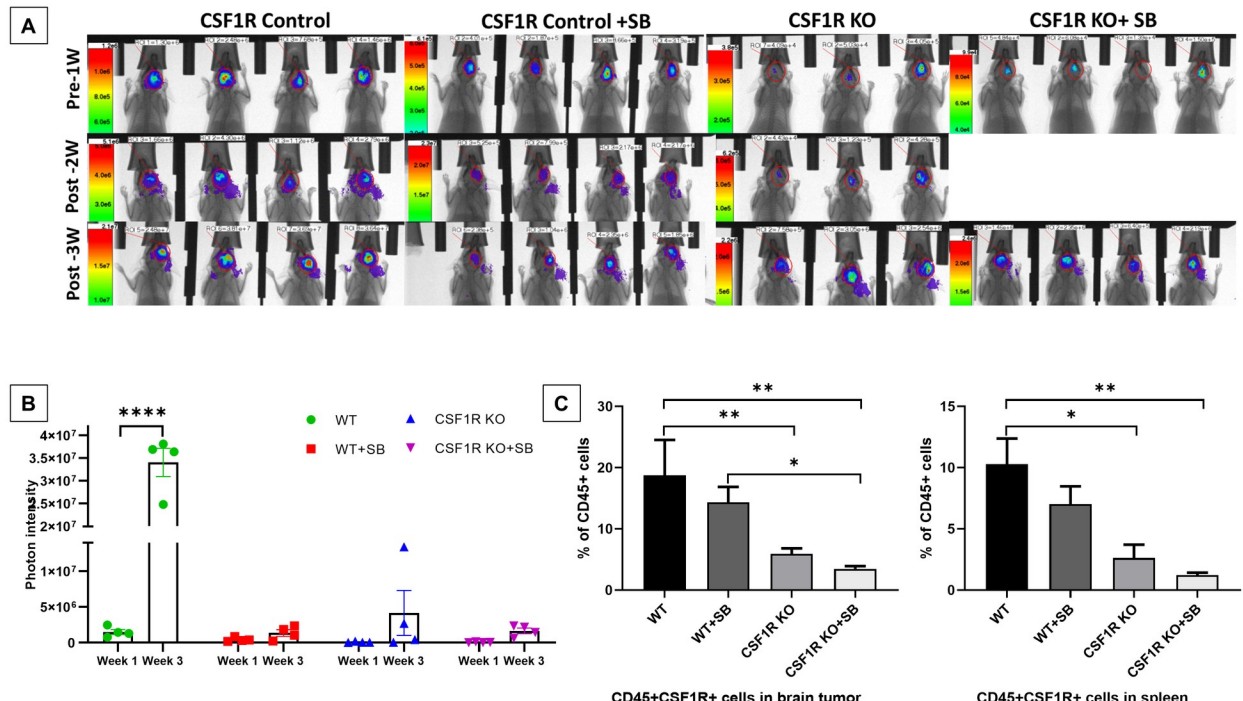

**Fig 3. (A and B)** Optical images and quantified photon intensities of pre and post-treatment (either vehicle or SB225002) showed significantly increased tumor growth in the vehicle-treated wild-type animals after 3 weeks. Knockout (KO) animals treated with either vehicle or SB225002 and wild-type animals treated with SB225002 (SB) did not show any significant tumor growth after 3 weeks. **(C)** Flow-cytometric analysis showing a significant decrease in CSF1R+ cells in brain tumor (left panel) and spleen (right panel) of the Knockout mice compared to the wild type mice treated with vehicle. Quantitative data are expressed in mean ± SEM. $^*P < .05$, $^{**}P < .01$, $^{****}P < .0001$. n = 4.

M2 macrophages significantly increased in tumors treated with TMZ (Figs 5 and 6). While treatment with anti-PD1, and HET0016 in wild type and KO mice increased the CD4+ and CD8+ T-cells most significantly. Representative dot-plots are provided in **S5–S7 Figs in** S1 File.

On the other hand, different cellular populations were significantly decreased in post-radiation tumors. All other treatments that targeted tumor-associated myeloid cells or checkpoint showed increased accumulation of CD4 and CD8 cells in the tumors but myeloid cell populations including MDSCs, CD11b+ cells, and macrophages showed insignificant changes in the TME compared to that of control tumors (Fig 6).

Then the photon intensity (intensity/sec/mm$^2$) was determined by making an irregular region of interest encircling the tumors at each time point. Fig 7 shows the tumor growth following different treatments. Tumors in all therapy groups except in Vatalanib treated animals, were stable following 1 week of treatments and there was no significant difference compared to that of vehicle-treated animals. However, Vatalanib treated animals showed significantly increased photon intensity indicating tumor growth following 1 week of treatments. Tumor growths were substantially increased in the vehicle, Vatalanib, and TMZ treated animals following 2 weeks of therapy indicating the development of resistance in the TMZ group. All other groups showed increased tumor growth but were significantly slower than that of vehicle, Vatalanib, or TMZ treated animals. It should be noted that the animals that received TME-associated cell-directed therapy showed significantly lower tumor growth 2 weeks following treatments. The animals that receive antiangiogenic (Vatalanib) and tumor cell-targeted (TMZ) therapy exhibited rebound tumor growth at 2 weeks of treatments.

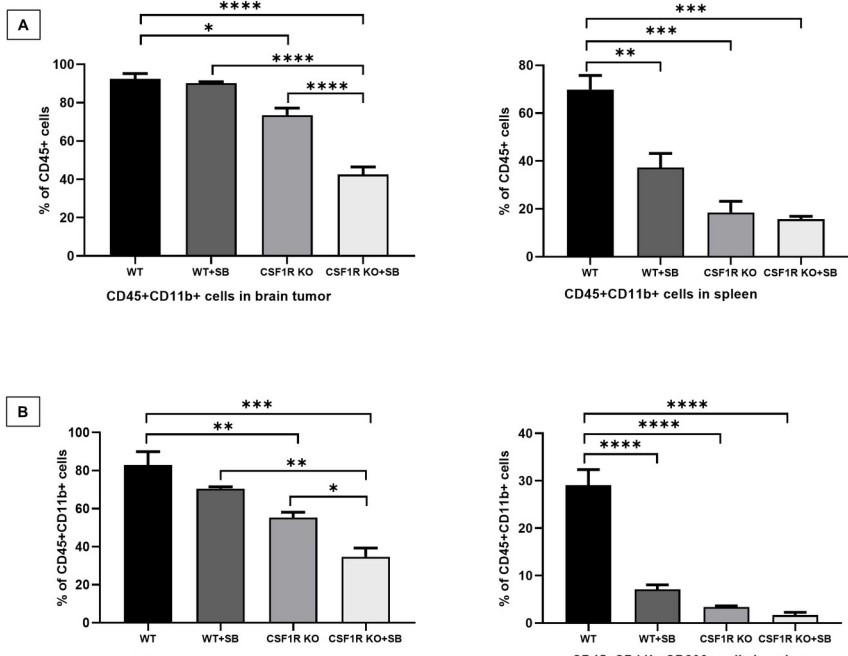

**Fig 4. Flowcytometric analysis of myeloid cell and tumor-associated macrophage populations in wild type and KO animals treated with either vehicle or SB225002 (SB).** Both (**A**) myeloid cells (CD45+CD11b+) and (**B**) TAMs (CD45+CD11b+CD206+) cells were significantly decreased in the brain (left panel) and spleen (right panel) of the KO mice while a decline was more prominent in the KO mice treated with SB225002. Quantitative data are expressed in mean ± SEM. *P < .05, **P < .01, ***P < .001, ****P < .0001. n = 3.

We instituted different treatments targeting both tumor cells and the tumor microenvironment including arachidonic acid metabolisms and anti-depressant (selective serotonin reuptake inhibitor (SSRI), fluoxetine) drugs alone or in combination with TMZ. Previously published studies showed improved survival with a 10mg/kg dose of HET0016 [8], therefore, to determine whether increasing the dose will increase the survival. A very high dose of HET0016 (50mg/kg/day) was also used. An analog of HET0016 was also used to see the effect on survival as the drug showed a similar effect *in vitro*. All treatments significantly increased the survival of animals bearing syngeneic GL261 GBM (Fig 8A). The most significantly increased survival was observed in animals' groups that were treated with TMZ, HET0016, TMZ+HET0016, and with a HET analog. Although Navarixin (IL-8CXCR2 axis blocker) increased the survival of the animals, the addition of TMZ did not improve survival (Fig 8B).

## Discussion

GBM is a devastating malignant tumor of the central nervous system. Once diagnosed the average survival is limited to15 months [87–90]. Currently, surgical resection followed by radiation and TMZ therapies is the standard of care for GBM patients [91]. With these extensive therapies, almost all patients show therapy resistance and recurrence of GBM [92]. To address resistance and recurrence, clinicians have adopted antiangiogenic therapies in recurrent GBM [93, 94]. These treatments decrease the formation of new blood vessels and decrease edema, thus reducing the dose of corticosteroids needed after therapy [95, 96]. Additionally, advanced immunotherapy and targeted therapies have been instituted [97]. However, early reports demonstrated that these are non-effective treatment strategies [10, 65, 93, 98–102]. Investigations

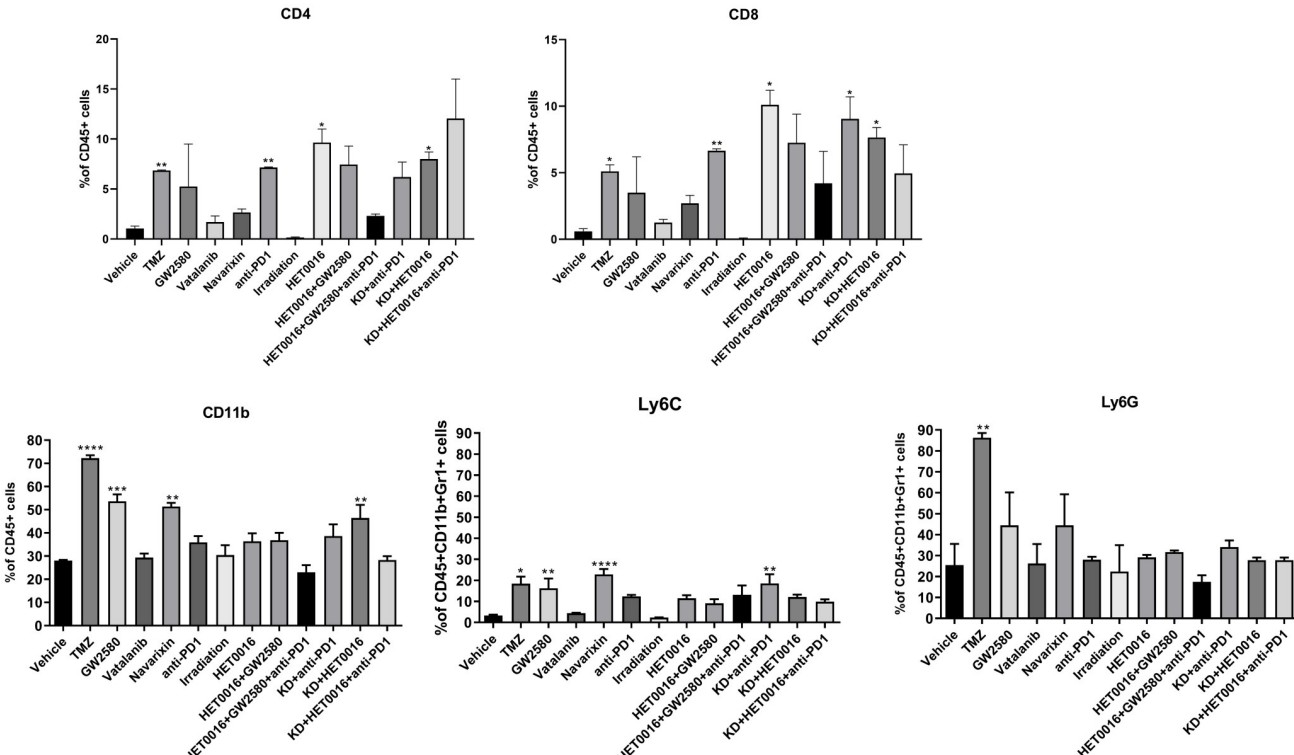

**Fig 5. Flow cytometric analysis of T-cells and myeloid cell populations in wild type and knockout animals.** There was a significant increase in CD4, CD8, CD11b, and Ly6G positive cells in tumors treated with TMZ, while irradiation caused a significant reduction in different cellular populations compared to the control group. All other treatments showed increased infiltration of CD4 and CD8 T-cells but insignificant changes in MDSCs, CD11b populations. Quantitative data presented in mean ± SEM. *P < .05, **P < .01, ***P < .001, ****P < .0001, n = 6. TMZ = temozolomide, PD 1 = programmed cells death protein 1, KD = Knockdown.

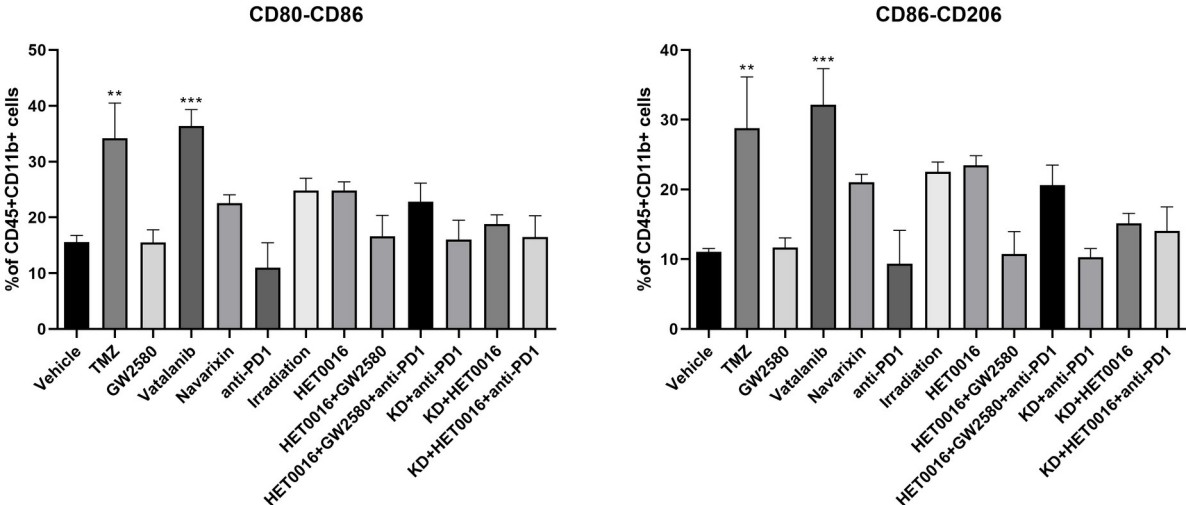

**Fig 6. Flowcytometric analysis of M1 and M2 macrophage populations.** Treatment with TMZ and Vatalanib increased the macrophage population significantly, and all other treatments changed the macrophage population inconsequentially. Quantitative data presented in mean ± SEM. **P < .01, ***P < .001, n = 6. TMZ = temozolomide, PD 1 = programmed cells death protein 1, KD = Knockdown.

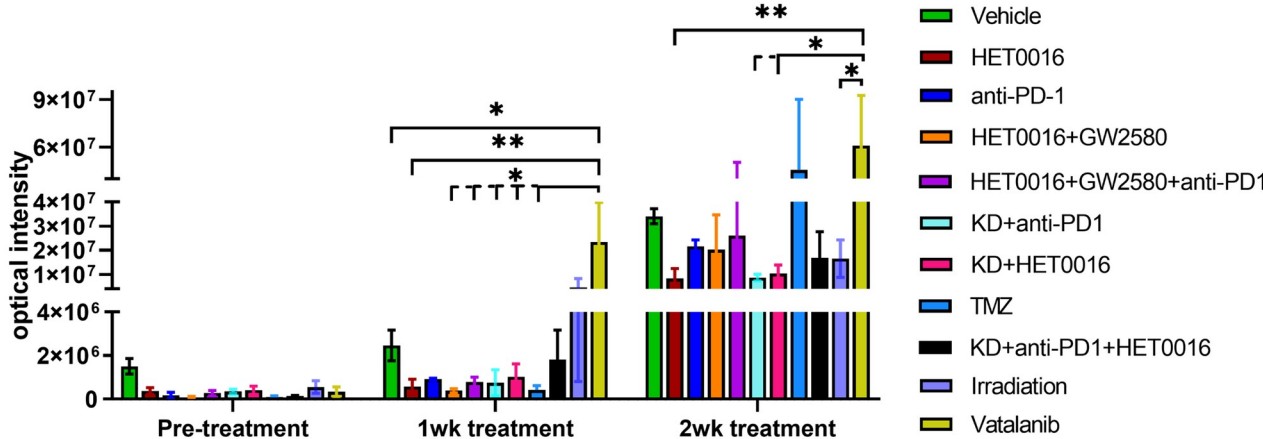

**Fig 7. Bioluminescent image-based analysis of tumor growth.** All animals underwent optical imaging to monitor tumor growth before starting the treatment (day 8 post-inoculation), 1 week, and 2 weeks after treatment. There was no significant difference between all treatment groups compared to that of vehicle-treated animals after 1 week of treatment except the Vatalanib treated group that showed significant tumor growth. Following 2 weeks of treatment, tumor growths were substantially increased in the vehicle, Vatalanib, and TMZ treated animals. All other groups showed increased tumor growth but were significantly slower than the above-mentioned groups. Quantitative data are expressed in mean ± SEM. *P < .05, **P < .01. n = 3.

from this lab indicated that most of the instituted therapies mobilized bone-marrow cells to the sites of GBM and orchestrated therapy resistance [10, 11]. The results showed that antiangiogenic therapies initiate alternate vascularization pathways and eventually increased neovascularization in therapy-resistant GBM [7, 65, 103]. Previous studies showed accumulation of angiogenic and vasculogenic myeloid cells at GBM sites following therapies [11, 102]. Furthermore, it is reported the process of vascular mimicry in which GBM cells transdifferentiate into glioma stem cells that can then form functional blood vessels [7, 63]. All of these results support the conclusion that the possible changes occurring in the TME following standard or investigational treatments in GBM have not been properly studied. This includes both changes in TME associated cells as well as the changes that occur in the metabolic cascade of TME

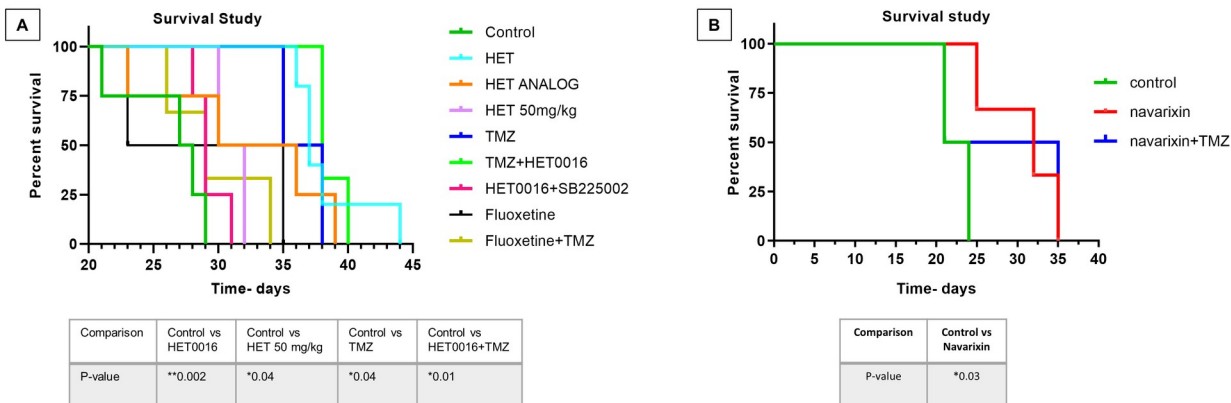

**Fig 8. Survival studies showing improved survival following the use of TME targeting agents. (A and B)** Kaplan-Meier curve showing significantly increased survival in animal groups treated with TMZ (50mg/kg/day, 3days/week), HET0016 (HET, 10mg/kg/day, 5days/week), TMZ+HET0016, and with a HET analog (10mg/kg/day, 5days/week). Although Navarixin (10mg/kg/day, 5 days/week) increased survival, the addition of TMZ with it did not improve the outcome. A Log-rank test (Mantel-Cox) was applied to determine the significance of differences among the groups. *P < .05, **P < .01. n = 3–5. Dose of Fluoxetine was 10 mg/kg/day, 3days/week. Dose of SB225002 was 10mg/kg/day, 5 days/week. Only wild type animals were used in survival studies.

associated cells. In this pilot study, these changes were investigated. To accomplish this, clinically practiced therapies (radiation and TMZ) as well as agents that targeted TME associated cells (CSF1R inhibitor GW2580 to target myeloid cells, IL-8-CXCR2 antagonists Navarixin and SB225002 to target stem cells causing vascular mimicry, an anti-PD1 antibody targeting immune suppressive molecules) and different metabolic pathways (HET0016 and its analog to target CYP4A-20-HETE axis of arachidonic acid metabolisms, fluoxetine to target serotonin reuptake) were used. Following therapies, the changes in the composition of TME-associated cells and the survival benefit of the therapeutic agents alone or in combination with TMZ were determined.

Our results clearly demonstrated the importance of TME associated CSF1R positive cells. Conditional knockout animals (CSF1R knockout) showed a decreased number of myeloid cells in the TME, whereas TMZ therapy increased the population of myeloid cells as well CD4 + and CD8+ T-cells in the treated GBM. Previous studies have shown the suppression of pro-inflammatory cytokine expression, activation of peripheral blood mononuclear cells (PBMC) or bone marrow-derived cells, and TME following TMZ therapy [104, 105]. The release of different chemotactic factors and cytokines allows bone marrow cells to mobilize and accumulate in the tumors [11, 65, 104]. The effect of Vatalanib in the mobilization and accumulation of bone marrow-derived cells in treated GBM has been published by us previously [11, 102, 103]. The current results are in corroboration with previous findings [11, 102, 103]. The previous studies showed the release of different chemoattractants in the GBM following Vatalanib, which initiated alternate neovascularization and development of resistance [65]. Previously, the reported results, as well as results from different investigators, have proven the importance of myeloid cells in developing therapy resistance in GBM and other cancers [11, 18, 20, 106–108]. Myeloid cells, such as macrophages and MDSCs, produce an immunosuppressive microenvironment that promotes tumor growth. Following chemotherapy, macrophage differentiation is altered to promote the production of cancer-supporting M2 macrophages in the TME [109]. Chemotherapy has also been shown to promote macrophage aggregation, thus facilitating cathepsin protease B- and S- mediated therapy resistance [110]. Some chemotherapeutic agents activate MDSCs to produce IL-1β. This leads to the secretion of IL-17 by CD4[+] T-cells [111]. Additionally, MDSCs have been shown to partake in the epithelial-mesenchymal transition, increase the production of multiple matrix metalloproteinases, and merge tumor cells [109–111]. Therefore, the addition of myeloid cell blockage could mitigate these mechanisms of resistance. The effect of CSF1R+ cells is also proven by the CSF1R+ conditional knockout mouse model in these current studies. However, it is to note that, previous investigations also indicated the development of resistance following long-term therapy using CSF1R inhibitors [112, 113]. This indicates the importance of sequential or intermittent therapy targeting GBM TME associated cells following or in between standard therapies for GBM.

To our surprise, a decreased accumulation of T-cells, as well as different myeloid cell populations in the TME following radiation therapies, was noted. This decreased accumulation of T-cells may be due to the disruption of intact blood vessels that act as a delivery system of T-cells to the tumor site. This disruption is likely caused by radiation therapy-induced necrosis in tumors leading to tumor cell death. Therefore, most tumor recurrence in post-radiation GBM occurs from the periphery of the irradiated areas where a few cells may have survived the radiation injury. In the previous publication [114], it is shown that sub-curative radiation increases tumor cell proliferation, migration, and invasion in a rat model of primary human GBM primarily by the increased expression level of MMP2, HIF-1α, and SDF-1α. In this study, a huge imbalance in T-lymphocyte and myeloid cells after 2 weeks following radiation therapy was observed. While T-lymphocytes are almost absent, CD11b positive myeloid cells (macrophages and MDSCs, in particular) were prevalent in the TME of post-radiation

animals. It is believed that this imbalance between the inflammatory cells following radiotherapy will make GBM recurrence inevitable by altering immune response and senescence-associated secretory profile (SASP) [115].

Our previous studies showed that the addition of HET0016 (blocker of CYP4A-20-HETE axis of arachidonic acid metabolisms) improved the survival of animals bearing patient-derived xenograft (PDX) GBM following 30Gy of radiotherapy [8]. HET0016 is known to inhibit tumor and endothelial cell (EC) proliferation, EC migration, and prevent neovascularization including vascular mimicry [47, 63, 64, 116]. Although the agents that prevent the repair of DNA damage have not been tested, the addition of a PARP inhibitor may also help prevent the recurrence of GBM following radiotherapy [117, 118]. However, in contrast to HET0016, the PARP inhibitor has a very narrow therapeutic window and causes severe toxicity [117]. Therefore, adding an inhibitor of arachidonic acid metabolic pathways may be useful in preventing the recurrence of post-radiation GBM. Results of survival studies using HET0016 showed encouraging results even with or without the addition of TMZ, which is in line with the previously published data [8]. As discussed above, TMZ therapy may cause an immunosuppressive environment and activation of PBMC, which eventually would allow the development of neovascularization and GBM recurrence. The addition of HET0016 would decrease tumor and endothelial cell proliferation, decrease the release of chemoattractant and vascularization factors, which will decrease GBM neovascularization and growth, and improve survival [8]. HET0016 could be a valid option for the treatment of GBM.

Previously, the effectiveness of HET0016 in controlling GBM and breast cancer have been reported by us [8, 47]. However, the presence of TME-associated cells following the treatment of HET0016 has not been reported yet. In this study, HET0016 treatment exhibited a similar phenomenon to that of myeloid cell-targeted therapies. It showed an increased T-cell population in the TME compared to that of vehicle and Vatalanib treated GBM. There was also a tendency to decrease immunosuppressive myeloid cell populations in the TME. Additionally, treatments using HET0016 and its analog showed significantly improved survival which corroborates with the previous reports [8]. The ongoing investigations show that the CYP4A-20-HETE pathway is active not only in tumor cells but also in TME associated myeloid cells. Inhibition of 20-HETE increases the cytotoxic T-cells population in *in vitro* studies (manuscript under preparation). Details of HET0016 mediated therapies and their mechanisms are discussed in the previous reports [8]. Although a few groups are studying the effects of CSF1R inhibitors (GW2580) in post-radiation models [119], none is working with 20-HETE inhibitors. It will be interesting to see which one (CSF1R inhibitors/HET0016) will have more efficiency to manipulate TME in post-radiation models for better outcomes. When compared the effect of HET0016 alone in wild-type animals with that of combination therapy of anti-PD-1 antibody in CSF1R knockout animals, there was no further decrease in the population of MDSC. However, addition of CSF1R inhibitor with anti-PD1 antibody in wild type animals further decrease MDCS (CD11b+ and Ly6G+ cells) compared to that of HET0016 alone. This indicate that the effect of CSF1R antagonist may have pronounced effects on all circulating CSF1R+ cells than the CSF1R knockout animals, where the conditional knockout animal showed 80% depletion of CSF1R+ cells. However, in GBM preclinical model (without radiation) better survival with HET0016 treatment alone compared to GW2580 or combination of GW2580 and HET0016 was observed. Therefore, we propose that the use of an inhibitor of the cytochrome P450 γ-hydroxylase pathway of arachidonic acid metabolisms may be used as an agent to target post-therapy GBM to prevent a recurrence or HET0016 can be used alone or combined with clinically practiced therapies to treat primary and recurrent GBM due to immunostimulatory effect of HET0016.

## Limitation and potential future experiments

In this study, immunohistochemical analysis was not included. The main aim was to determine the different cellular populations in TME by flow cytometry. Because of the smaller tumor size following different therapies that decreased the tumor tremendously, there was no tissue left for histochemistry. In the future, it is planned to have cohorts for the collection of tissue for immunohistochemistry. Further studies are being conducted to analyze TME to correlate with immunohistochemistry and proteomics for GBM and breast cancers. In the current studies, TME-associated cells at an earlier time point were not determined. In the future, systematic investigations will be conducted to determine the composition of different bone marrow-derived cells in GBM TME in respect of different populations of macrophages, microglia, MDSCs, and T-cells. Also, the lack of analysis of TME-associated cells in animals treated with different agents that went for survival is a limitation of this study. The groups of animals treated with TMZ+ HET0016 and HET0016+SB225002 were only for survival but BLI was missed. Results of the TMZ+HET0016 group in different animal models have already been published by us previously using MRI [8]. Now we are working with different CXCR2 antagonists with HET0016 and the results will be published once the studies are done. We have not used all the treatments and survival studies in CSF1R KO animals. In the future, a systematic investigation will be conducted using CSF1R KO animals.

## In conclusion

Current GBM therapies should be revisited to add agents to prevent the accumulation of bone marrow-derived cells in the TME or to prevent the effect of immune-suppressive myeloid cells in causing alternative neovascularization, the revival of glioma stem cells, and recurrence. Instead of concurrent therapy, a sequential strategy would be best to target TME associated cells.

## Supporting information

**S1 File.**
(DOCX)

## Acknowledgments

The authors like to acknowledge the help of the core facility of small animal imaging (CIFSA) for acquiring optical images.

## Author Contributions

**Conceptualization:** Bhagelu R. Achyut, Ali Syed Arbab.

**Data curation:** Sehar Ali, Roxan Ara, Kartik Angara, Bhagelu R. Achyut, Mohammad H. Rashid.

**Formal analysis:** Sehar Ali, Thaiz F. Borin, Raziye Piranlioglu, Kartik Angara, Bhagelu R. Achyut, Ali Syed Arbab, Mohammad H. Rashid.

**Funding acquisition:** Ali Syed Arbab.

**Investigation:** Thaiz F. Borin, Roxan Ara, Kartik Angara, Bhagelu R. Achyut, Ali Syed Arbab, Mohammad H. Rashid.

**Methodology:** Sehar Ali, Thaiz F. Borin, Kartik Angara, Bhagelu R. Achyut, Ali Syed Arbab, Mohammad H. Rashid.

**Project administration:** Ali Syed Arbab.

**Resources:** Iryna Lebedyeva, Ali Syed Arbab.

**Supervision:** Thaiz F. Borin, Ali Syed Arbab.

**Validation:** Thaiz F. Borin, Raziye Piranlioglu.

**Writing – original draft:** Mohammad H. Rashid.

**Writing – review & editing:** Raziye Piranlioglu, Bhagelu R. Achyut, Ali Syed Arbab, Mohammad H. Rashid.

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
