## [Decision Letter · Decision Letter 0]

8 Dec 2020

PONE-D-20-32987

Dear Editor,

Changes in the tumor microenvironment and outcome for TME-targeting therapy in glioblastoma: A pilot study

PLOS ONE

Dear Dr. Arbab,

Thank you for submitting your manuscript to PLOS ONE. After careful consideration, we feel that it has merit but does not fully meet PLOS ONE’s publication criteria as it currently stands. Therefore, we invite you to submit a revised version of the manuscript that addresses the points raised during the review process.

We look forward to receiving your revised manuscript.

Kind regards,

Ilya Ulasov, Ph.D

Academic Editor

PLOS ONE

Journal Requirements:

2. Please note that PLOS does not permit references to “data not shown.” Authors should provide the relevant data within the manuscript, the Supporting Information files, or in a public repository. If the data are not a core part of the research study being presented, we ask that authors remove any references to these data.

3. During your revisions, please note that a simple title correction is required: Please remove "Dear Editor," from the online submission form. Please ensure this is updated in the manuscript file and the online submission information.

4. Please include a brief description or suitable reference for the preparation of HET0016 including reagents and synthesis conditions in your Methods.

5.Please provide the product number and any lot numbers of the antibodies purchased for your study.

6. Please ensure you have included the strain and source for all the mice included in your study.

7. In your Methods section, please provide additional details regarding the cell lines used in your study and ensure you have described the source. Please provide additional information about each of the cell lines used in this work, including any quality control testing procedures (authentication, characterisation, and mycoplasma testing).

For more information regarding PLOS' policy on materials sharing and reporting, see https://journals.plos.org/plosone/s/materials-and-software-sharing#loc-sharing-materials, and for more information on PLOS ONE's guidelines for research using cell lines, see https://journals.plos.org/plosone/s/submission-guidelines#loc-cell-lines.

8.PLOS ONE now requires that authors provide the original uncropped and unadjusted images underlying all blot or gel results reported in a submission’s figures or Supporting Information files. This policy and the journal’s other requirements for blot/gel reporting and figure preparation are described in detail at https://journals.plos.org/plosone/s/figures#loc-blot-and-gel-reporting-requirements and https://journals.plos.org/plosone/s/figures#loc-preparing-figures-from-image-files. When you submit your revised manuscript, please ensure that your figures adhere fully to these guidelines and provide the original underlying images for all blot or gel data reported in your submission. See the following link for instructions on providing the original image data: https://journals.plos.org/plosone/s/figures#loc-original-images-for-blots-and-gels.

Reviewers' comments:

Reviewer's Responses to Questions

**Comments to the Author**

1. Is the manuscript technically sound, and do the data support the conclusions?

Reviewer #1: Yes

Reviewer #2: No

2. Has the statistical analysis been performed appropriately and rigorously? 

Reviewer #1: Yes

Reviewer #2: I Don't Know

3. Have the authors made all data underlying the findings in their manuscript fully available?

Reviewer #1: Yes

Reviewer #2: No

4. Is the manuscript presented in an intelligible fashion and written in standard English?

Reviewer #1: Yes

Reviewer #2: Yes

5. Review Comments to the Author

Reviewer #1: English can improve further, but other than that the work is done very well. Detail of shortcomings is also included. They have used majorly surface markers to analyze the TME rather than more elaborate settings including cytokines. But this is also well explained.I recommend publication.

Reviewer #2: In the present manuscript the authors present a pilot study addressing the changes in immune cell populations of the TME of a syngeneic mouse model of GBM, in relation to the different treatments administered to recapitulate standard and novel candidate therapies for human GBM. They also measure tumor growth and survival following different treatments. In some experiments they use CSF1R KO animals to address the involvement of tumor-promoting myeloid cells. Their aim is to show associations between changes in the TME-associated cells and the use of the distinct therapeutic agents. The accumulation of bone marrow-derived cells in the GBM TME and the effect of immune-suppressive myeloid cells in causing alternative neovascularization, the revival of glioma stem cells, and recurrence is already know and well-documented. The percentages of immune cells in GBM TME have not been addressed in relation to GBM therapies before and their characterization would be beneficial for a translational application in GBM cure; yet, as it is now, the study is too preliminary for publication and requires further integrations and some corrections to enforce the solidity and informativeness of data before being accepted. The authors themselves indicate the limitations of their study and point at future experiments to be done (lines 569-585). Immunohistochemical analyses might be postponed to future characterizations, but cytofluorimetric analyses in the present study are not exhaustive and need to be revised.

No details to the antibodies used are reported in the M&M section (clone, fluorochrome, vendor); a live/dead staining should have been included, to avoid to count dead cells in the acquisition. It is not clear how many surface markers were studied within each multicolor staining panels, the latter information is not provided and the exact gating strategies are not shown per each analysis either. Without the use of multicolor staining panel it is not possible to univocally identify/distinguish the immune population of interest. It is well-known that the phenotypic distinction between M1 and M2 macrophages is not clear-cut and most large scale transcriptome analyses show that macrophages have a mixed phenotype expressing both M1 and M2 markers: therefore, the couples of markers used herein appear not sufficient; in line with this, the reciprocal percentages are almost the same. On the other hand, among TAMs, it should be important to dissect the proportions of peripherally derived infiltrating BMDMs versus resident microglia, according to already published cytofluorimetric methods that rely on differential cell-surface expression of CD45 and CD11b by these two cell types (CD45loCD11b+ defining putative microglia, CD45hiCD11b+ defining putative BMDMs). This might help explain results of treatments in Fig. 3. Multicolor flow cytometry panels to comprehensively immunoprofile TME-associated cells, should include different markers for myeloid cells (CD11b, Gr1, Ly6G, Ly6C, CD11c, Tie2, and MHC class II), for lymphoid cells (CD3, CD4, CD8 and possibly CD19) and possibly activation status and PD1 expression on T cells (as anti-PD1 therapy is evaluated as well).

Stainings with isotype control Abs should be done, resulting useful especially for myeloid cells. Plots are not indicated for each figure, for each condition tested and for each control. The cytofluorimetric plots provided do not seem to be correctly compensated and often many events are found in the diagonal, as if they were dead cells. The Ab used to detect CSFR gives a very low staining, so it is very difficult to appreciate positive cells (Suppl fig 2: all plots for all mice should be shown, including plots of wt mice, where CD45+CSFR1+ cells are present).

Compensation and gating might be checked and consequently the percentages should be recalculated. Some of them already appear overestimated (e.g., in Fig. 5, the sum of CD45+CD4+, CD45+CD8+ and CD45+CD11b+ cells in the condition TMZ is far over 100%….). Gating strategies and plots should be always shown (they are not shown for related graphs in Fig. 5 and 6) for clarity. Graphs of cumulative percentages of different CD45+ subpopulations for each condition could be less ambiguous.

The authors should better detail figure legends: every legend should include the timepoint of assay, the number of mice/group and the number of repeated measures. The number of mice (n) is not always indicated and sometimes there are discrepancies (Fig1B legend: n=8 and not 10; Fig 3: the number of dots shown in panel B do not match to the number of mice shown in panel A for the corresponding condition).

The tumor growth and survival data are not measured for the same treatments analyzed for TME-associated cells. A systematic investigation should be conducted, including CSF1R KO animals, as also said by the authors. For the tumor mass studies, too few mice are used per group (n=3), resulting in high SEM and impeding the identification of significant differences among treatments. More importantly, not all the combinations of treatments have been addressed, therefore controls are missing. Radiation plus TMZ plus/minus antiangiogenic drug is missing as well, as a reference of standard therapies for patients. Currently, surgical resection followed by radiation and TMZ therapies is the standard of care for GBM patients.

Further on, the observation that “the animals that received TME-associated cell-directed therapy showed significantly lower tumor growth 2 weeks following treatments” is correct compared to the condition “vehicle” (=no treatment), but not compared to radiation, that exerts a similar effect (Fig. 7), arguing the advantage of preferring selective TME-associated cell-directed therapy. Finally one of the conclusion “instead of concurrent therapy, a sequential strategy would be best to target TME associated cells” is not really sustained by data, to prove this the authors should have included sequential administration of different treatments in their experimental setting.

Some figures could be pooled together; the introduction and the discussion are quite long compared to results; the discussion is too speculative.

6. PLOS authors have the option to publish the peer review history of their article (what does this mean?). If published, this will include your full peer review and any attached files.

Reviewer #1: No

Reviewer #2: No

---

## [Author Response · Author response to Decision Letter 0]

12 Jan 2021

Journal Requirements:

Answer: Thank you. We have followed the style requirement and format

2. Please note that PLOS does not permit references to “data not shown.” Authors should provide the relevant data within the manuscript, the Supporting Information files, or in a public repository. If the data are not a core part of the research study being presented, we ask that authors remove any references to these data.

Answer: Thank you. We have omitted the term and rephrased the sentences in the revised manuscript

3. During your revisions, please note that a simple title correction is required: Please remove "Dear Editor," from the online submission form. Please ensure this is updated in the manuscript file and the online submission information.

Answer: Corrected

4. Please include a brief description or suitable reference for the preparation of HET0016 including reagents and synthesis conditions in your Methods.

 Answer: We have included the relevant reference (Menu’s paper). We have also included the synthesis strategies for HET0016 and its analog in the revised manuscript as supplemental/supporting information.

5.Please provide the product number and any lot numbers of the antibodies purchased for your study.

 Answer: We have provided the requested information. 

6. Please ensure you have included the strain and source for all the mice included in your study.

Answer: We have already provided all the necessary information already. We have included C57/Bl6 purchased from Jackson.

7. In your Methods section, please provide additional details regarding the cell lines used in your study and ensure you have described the source. Please provide additional information about each of the cell lines used in this work, including any quality control testing procedures (authentication, characterization, and mycoplasma testing).

For more information regarding PLOS' policy on materials sharing and reporting, see https://journals.plos.org/plosone/s/materials-and-software-sharing#loc-sharing-materials, and for more information on PLOS ONE's guidelines for research using cell lines, see https://journals.plos.org/plosone/s/submission-guidelines#loc-cell-lines.

Answer: We have received GL261 mouse GBM cells from repository of NCI. The cells are routinely cultured and propagated in our laboratory. Mycoplasma is routinely checked and the cells are treated with standard antibiotics during culture. This study was started in later part of 2017. GL261 was authenticated using short tandem repeat (STR) profiling. The information has been added in the revised manuscript

 8.PLOS ONE now requires that authors provide the original uncropped and unadjusted images underlying all blot or gel results reported in a submission’s figures or Supporting Information files. This policy and the journal’s other requirements for blot/gel reporting and figure preparation are described in detail at https://journals.plos.org/plosone/s/figures#loc-blot-and-gel-reporting-requirements and https://journals.plos.org/plosone/s/figures#loc-preparing-figures-from-image-files. When you submit your revised manuscript, please ensure that your figures adhere fully to these guidelines and provide the original underlying images for all blot or gel data reported in your submission. See the following link for instructions on providing the original image data: https://journals.plos.org/plosone/s/figures#loc-original-images-for-blots-and-gels.

Answer: We have provided the original blot images in the supplemental/supporting information documents

Answer: It is added in the manuscript as supplemental/supporting information

 

Reviewer #1: English can improve further, but other than that the work is done very well. Detail of shortcomings is also included. They have used majorly surface markers to analyze the TME rather than more elaborate settings including cytokines. But this is also well explained. I recommend publication.

Answer: Thank you for your kind comments. This is a pilot study where we wanted to see the changes in the surface marker in the TME associated cells following different therapies. Our strategy was to debilitate the CD45+ cells and its different lineages such as lymphocytes, macrophages, and MDSC. We have tried to correct the English.

Reviewer #2: In the present manuscript the authors present a pilot study addressing the changes in immune cell populations of the TME of a syngeneic mouse model of GBM, in relation to the different treatments administered to recapitulate standard and novel candidate therapies for human GBM. They also measure tumor growth and survival following different treatments. In some experiments they use CSF1R KO animals to address the involvement of tumor-promoting myeloid cells. Their aim is to show associations between changes in the TME-associated cells and the use of the distinct therapeutic agents. The accumulation of bone marrow-derived cells in the GBM TME and the effect of immune-suppressive myeloid cells in causing alternative neovascularization, the revival of glioma stem cells, and recurrence is already know and well-documented. The percentages of immune cells in GBM TME have not been addressed in relation to GBM therapies before and their characterization would be beneficial for a translational application in GBM cure; yet, as it is now, the study is too preliminary for publication and requires further integrations and some corrections to enforce the solidity and informativeness of data before being accepted. The authors themselves indicate the limitations of their study and point at future experiments to be done (lines 569-585). Immunohistochemical analyses might be postponed to future characterizations, but cytofluorimetric analyses in the present study are not exhaustive and need to be revised.

Answer: Thank you for the in depth analysis and review of our manuscript and suggestions. As we have mentioned in our manuscript that this is a pilot study and our main aims were to determine the changes in the TME associated CD45+ cells. We also mentioned the shortcomings and future experimental direction to investigate in details of the studies related to TME changes.

No details to the antibodies used are reported in the M&M section (clone, fluorochrome, vendor); a live/dead staining should have been included, to avoid to count dead cells in the acquisition. It is not clear how many surface markers were studied within each multicolor staining panels, the latter information is not provided and the exact gating strategies are not shown per each analysis either. Without the use of multicolor staining panel it is not possible to univocally identify/distinguish the immune population of interest. It is well-known that the phenotypic distinction between M1 and M2 macrophages is not clear-cut and most large scale transcriptome analyses show that macrophages have a mixed phenotype expressing both M1 and M2 markers: therefore, the couples of markers used herein appear not sufficient; in line with this, the reciprocal percentages are almost the same. On the other hand, among TAMs, it should be important to dissect the proportions of peripherally derived infiltrating BMDMs versus resident microglia, according to already published cytofluorimetric methods that rely on differential cell-surface expression of CD45 and CD11b by these two cell types (CD45loCD11b+ defining putative microglia, CD45hiCD11b+ defining putative BMDMs). This might help explain results of treatments in Fig. 3. Multicolor flow cytometry panels to comprehensively immunoprofile TME-associated cells, should include different markers for myeloid cells (CD11b, Gr1, Ly6G, Ly6C, CD11c, Tie2, and MHC class II), for lymphoid cells (CD3, CD4, CD8 and possibly CD19) and possibly activation status and PD1 expression on T cells (as anti-PD1 therapy is evaluated as well).

Answer: Thank you for your comments. As we have mentioned this is a pilot study therefore suggested detailed analysis is not possible at this moment with the existing data. Please see the panels of markers and fluorochromes that we used to analyze the TME associated cells that are CD45+. CD45+ low or high is a subjective analysis. Our lab do not differentiate microglia based on the CD45+low profile. We do not even use any known marker for macrophages as marker of microglia. Our microglia marker are CD11b+/Tmem119+/P2RY12+ cells. In this study, we have not analyze the proportion of microglia. Our main aim was to determine the population of bone marrow derived cells following different therapies in GBM. We have reported the significant findings although we have used different cellular profiles.

We have used Accuri C6 flow machine and its physics is different from BD based flow cytometry (which has vacuum based fluidics and PMT based adjustment of cellular/fluorophore population). Our gating strategies were to gate on CD45+ cells in the single cell suspension from the brain tumors and then adjust based on the positivity of different markers. We have corrected our images to show the correct population. Because of large number of samples for the analysis of flow, our laboratory’s standard practice is to fix the cells with 3% paraformaldehyde as soon as we make single cell suspension. Therefore, chance of dead cell in the population is little. However, we checked the dead cells randomly by staining with 7AAD.

 FITC PE PerCP or PERCP-Cy5.5 APC

TUBE 1 45 11b ly6c gr1

TUBE 2 gr1 45 11b ly6g

TUBE 3 45 80 86 206

TUBE 4 4 45 25 279

tube 5 8 45 62l 279

Tube 6 45 133 - 44

Stainings with isotype control Abs should be done, resulting useful especially for myeloid cells. Plots are not indicated for each figure, for each condition tested and for each control. The cytofluorimetric plots provided do not seem to be correctly compensated and often many events are found in the diagonal, as if they were dead cells. The Ab used to detect CSFR gives a very low staining, so it is very difficult to appreciate positive cells (Suppl fig 2: all plots for all mice should be shown, including plots of wt mice, where CD45+CSFR1+ cells are present).

Answer: Our strategies were to make 5 tubes for compensation; unstained, and stained with CD45-FITC, CD45-PE, CD45-PerCP and CD45-APC. Data from these 5 tubes will allow us to properly delineate the fluorochrome and their compensation in Accuri C6 flow. Our practical experience showed that isotype control for each antibody types tagged with corresponding fluorochrome sometimes showed non-specific binding especially in myeloid cells unless we block them with Fc-blocker. We use Fc-blocker in each tube and used single colored CD45+ cells to make proper compensation and gating strategies. 

Compensation and gating might be checked and consequently the percentages should be recalculated. Some of them already appear overestimated (e.g., in Fig. 5, the sum of CD45+CD4+, CD45+CD8+ and CD45+CD11b+ cells in the condition TMZ is far over 100%….). Gating strategies and plots should be always shown (they are not shown for related graphs in Fig. 5 and 6) for clarity. Graphs of cumulative percentages of different CD45+ subpopulations for each condition could be less ambiguous.

Answer: We have included the gating strategies of cells for figure 5 and 6 in the revised supplemental/supporting information. For figure 5, tubes 1, 2, 4 and 5 were used (see above) and for Figure 6, tube 3 was used. We have corrected the axis to show the percentage related to our gating strategies.

The authors should better detail figure legends: every legend should include the timepoint of assay, the number of mice/group and the number of repeated measures. The number of mice (n) is not always indicated and sometimes there are discrepancies (Fig1B legend: n=8 and not 10; Fig 3: the number of dots shown in panel B do not match to the number of mice shown in panel A for the corresponding condition).

Answer: We have corrected as suggested. Thank you pointing out Fig 3. The dot point was cut off when graph height was adjusted. Now it is corrected.

The tumor growth and survival data are not measured for the same treatments analyzed for TME-associated cells. A systematic investigation should be conducted, including CSF1R KO animals, as also said by the authors. For the tumor mass studies, too few mice are used per group (n=3), resulting in high SEM and impeding the identification of significant differences among treatments. More importantly, not all the combinations of treatments have been addressed, therefore controls are missing. Radiation plus TMZ plus/minus antiangiogenic drug is missing as well, as a reference of standard therapies for patients. Currently, surgical resection followed by radiation and TMZ therapies is the standard of care for GBM patients.

Answer: The reviewer is asking full-blown studies for 5 years’ R01 type proposal, which need more work force and fund, which are limited now in my lab. As we mentioned this is a pilot study and we said the further studies would be done in future. Systematic investigations needs more animals and time. We do not know there is any model of surgical resection of GBM in mouse. We will appreciate the help from the reviewer.

Further on, the observation that “the animals that received TME-associated cell-directed therapy showed significantly lower tumor growth 2 weeks following treatments” is correct compared to the condition “vehicle” (=no treatment), but not compared to radiation, that exerts a similar effect (Fig. 7), arguing the advantage of preferring selective TME-associated cell-directed therapy. Finally one of the conclusion “instead of concurrent therapy, a sequential strategy would be best to target TME associated cells” is not really sustained by data, to prove this the authors should have included sequential administration of different treatments in their experimental setting.

Answer: Thank you for your comment. We will consider your suggestion in our future studies. Our ongoing studies indicating that concurrent therapy is failing.

Some figures could be pooled together; the introduction and the discussion are quite long compared to results; the discussion is too speculative.

Answer: Thank you for your suggestion.

---

## [Editor Report · Decision Letter 1]

25 Jan 2021

Changes in the tumor microenvironment and outcome for TME-targeting therapy in glioblastoma: A pilot study

PONE-D-20-32987R1

Dear Dr. Arbab,

We’re pleased to inform you that your manuscript has been judged scientifically suitable for publication and will be formally accepted for publication once it meets all outstanding technical requirements.

Kind regards,

Ilya Ulasov, Ph.D

Academic Editor

PLOS ONE

---

## [Editor Report · Acceptance letter]

27 Jan 2021

PONE-D-20-32987R1 

Changes in the tumor microenvironment and outcome for TME-targeting therapy in glioblastoma: A pilot study 

Dear Dr. Arbab:

I'm pleased to inform you that your manuscript has been deemed suitable for publication in PLOS ONE. Congratulations! Your manuscript is now with our production department. 

Kind regards, 

on behalf of

Dr. Ilya Ulasov 

Academic Editor

PLOS ONE